# GNNEvaluator: Evaluating GNN Performance On Unseen Graphs Without Labels

**Xin Zheng**[1], **Miao Zhang**[2], **Chunyang Chen**[1], **Soheila Molaei**[3], **Chuan Zhou**[4], **Shirui Pan**[5]*

[1]Monash University, Australia,  [2]Harbin Institute of Technology (Shenzhen), China
[3]University of Oxford, UK,  [4]Chinese Academy of Sciences, China,  [5]Griffith University, Australia
xin.zheng@monash.edu, zhangmiao@hit.edu.cn, chunyang.chen@monash.edu
soheila.molaei@eng.ox.ac.uk, zhouchuan@amss.ac.cn, s.pan@griffith.edu.au

## Abstract

Evaluating the performance of graph neural networks (GNNs) is an essential task for practical GNN model deployment and serving, as deployed GNNs face significant performance uncertainty when inferring on unseen and unlabeled test graphs, due to mismatched training-test graph distributions. In this paper, we study a *new* problem, **GNN model evaluation**, that aims to assess the performance of a specific GNN model trained on labeled and observed graphs, by precisely estimating its performance (*e.g.*, node classification accuracy) on unseen graphs without labels. Concretely, we propose a two-stage GNN model evaluation framework, including (1) DiscGraph set construction and (2) `GNNEvaluator` training and inference. The DiscGraph set captures wide-range and diverse graph data distribution discrepancies through a discrepancy measurement function, which exploits the outputs of GNNs related to latent node embeddings and node class predictions. Under the effective training supervision from the DiscGraph set, `GNNEvaluator` learns to precisely estimate node classification accuracy of the to-be-evaluated GNN model and makes an accurate inference for evaluating GNN model performance. Extensive experiments on real-world unseen and unlabeled test graphs demonstrate the effectiveness of our proposed method for GNN model evaluation.

## 1 Introduction

As prevalent graph data learning models, graph neural networks (GNNs) have attracted much attention and achieved great success for various graph structural data related applications in the real world [36, 38, 35, 41, 42, 14, 21, 46, 44]. In practical scenarios of GNN model deployment and serving [37, 40, 43], understanding and evaluating GNN models' performance is a vital step [3, 13, 22], where model designers need to determine if well-trained GNN models will perform well in practical serving, and users want to know how the in-service GNN models will perform when inferring on their own test graphs [10]. Conventionally, model evaluation utilizes well-annotated datasets for testing to calculate certain model performance metrics (*e.g.*, accuracy and F1-score) [3, 24, 23, 20]. However, it may fail to work well in real-world GNN model deployment and serving, where the unseen test graphs are usually not annotated, making it difficult to obtain essential model performance metrics without ground-truth labels [4, 5]. As shown in Fig. 1 (a-1), taking *node classification accuracy* metric as an example, it is typically calculated as the percentage of correctly predicted node labels. However, when ground-truth node class labels are unavailable, we can not verify whether the predictions of GNNs are correct, and thus cannot get the overall accuracy of the model.

In light of this, a natural question, referred to **GNN model evaluation** problem, arises: *in the absence of labels in an unseen test graph, can we estimate the performance of a well-trained GNN model?*

---

*Corresponding author

37th Conference on Neural Information Processing Systems (NeurIPS 2023).

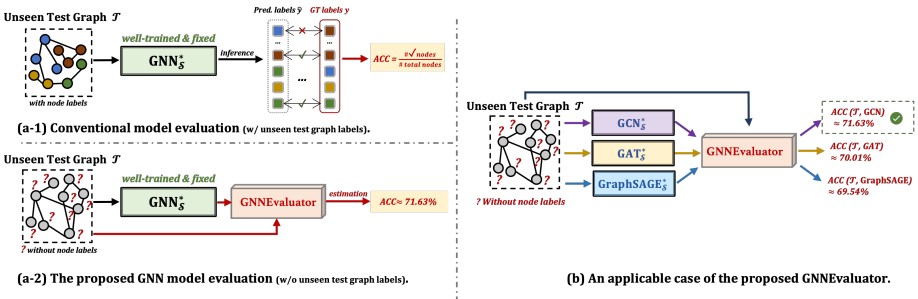

Figure 1: Illustration of conventional model evaluation *vs.* the proposed GNN model evaluation with its applicable case. The symbol $*$ indicates GNNs are well-trained and fixed during model evaluation.

In this work, we provide a confirmed answer together with an effective solution to this question, as shown in Fig.1 (a-2). Given a well-trained GNN model and an unseen test graph without labels, GNN model evaluation directly outputs the overall accuracy of this GNN model. This enables users to understand their GNN models at hand, benefiting many GNN deployment and serving scenarios in the real world [2, 8, 30, 25, 47, 45, 16, 15]. For example, given a GNN model in service (*e.g.*, a GNN model trained and served with Amazon DGL and SageMaker [1]), GNN model evaluation can provide users with a confidence score based on its estimated node classification accuracy, so that users know how much faith they should place in GNN-predicted node labels on their own unlabeled test graphs. Moreover, as shown in Fig. 1 (b), given a set of available well-trained GNNs in service, GNN model evaluation can provide users with reliable guidance to make an informed choice among these deployed GNN models on their own unlabeled test graphs.

In this paper, we focus on the node classification task, with accuracy as the primary GNN model evaluation metric. Developing effective GNN model evaluation methods for this task faces three-fold challenges: **Challenge-1:** The distribution discrepancies between various real-world unseen test graphs and the observed training graph are usually complex and diverse, incurring significant uncertainty for GNN model evaluation. **Challenge-2:** It is not allowed to re-train or fine-tune the practically in-service GNN model to be evaluated, and the only accessible model-related information is its outputs. Hence, how to fully exploit the limited GNN outputs and integrate various training-test graph distribution differences into discriminative discrepancy representations is critically important. **Challenge-3:** Given the discriminative discrepancy representations of training-test graph distributions, how to develop an accurate GNN model evaluator to estimate the node classification accuracy of an in-service GNN on the unseen test graph is the key to GNN model evaluation.

To address the above challenges, in this work, we propose a two-stage GNN model evaluation framework, including (1) DiscGraph set construction and (2) `GNNEvaluator` training and inference. More specifically, in the first stage, we first derive a set of meta-graphs from the observed training graph, which involves wide-range and diverse graph data distributions to simulate (ideally) any potential unseen test graphs in practice, so that the complex and diverse training-test graph data distribution discrepancies can be effectively captured and modeled (*Challenge-1*). Based on the meta-graph set, we build a set of discrepancy meta-graphs (DiscGraph) by jointly exploiting latent node embeddings and node class predictions of the GNN with a discrepancy measurement function, which comprehensively integrates GNN output discrepancies to expressive graph structural discrepancy representations involving discrepancy node attributes, graph structures, and accuracy labels (*Challenge-2*). In the second stage, we develop a `GNNEvaluator` composed of a typical GCN architecture and an accuracy regression layer, and we train it to precisely estimate node classification accuracy with effective supervision from the representative DiscGraph set (*Challenge-3*). During the inference, the trained `GNNEvaluator` could directly output the node classification accuracy of the in-service GNN on the unseen test graph without any node class labels.

In summary, the contributions of our work are listed as follows:

• **Problem.** We study a new research problem, GNN model evaluation, which opens the door for understanding and evaluating the performance of well-trained GNNs on unseen real-world graphs without labels in practical GNN model deployment and serving.

- **Solution.** We design an effective solution to simulate and capture the discrepancies of diverse graph data distributions, together with a `GNNEvaluator` to estimate node classification accuracy of the in-service GNNs, enabling accurate GNN model evaluation.
- **Evaluation.** We evaluate our method on real-world unseen and unlabeled test graphs, achieving a low error (*e.g.*, as small as 2.46%) compared with ground-truth accuracy, demonstrating the effectiveness of our method for GNN model evaluation.

**Prior Works**. Our research is related to existing studies on *predicting model generalization error*, which aims to estimate a model's performance on unlabeled data from the unknown and shifted distributions [6, 9, 34, 10, 4]. However, these researches are designed for data in Euclidean space (*e.g.*, images) while our research is dedicatedly designed for graph structural data. Our research also significantly differs from others in unsupervised graph domain adaption [33, 39, 29], out-of-distribution (OOD) generalization [18, 48], and OOD detection [19, 26], in the sense that we aim to estimate well-trained GNN models' performance, rather than improve the generalization ability of new GNN models. Detailed related work can be found in Appendix A.

## 2   Problem Definition

**Preliminary.** Consider that we have a fully-observed training graph $\mathcal{S} = (\mathbf{X}, \mathbf{A}, \mathbf{Y})$, where $\mathbf{X} \in \mathbb{R}^{N \times d}$ denotes $N$ nodes with $d$-dimensional features, $\mathbf{A} \in \mathbb{R}^{N \times N}$ denotes the adjacency matrix indicating the edge connections, and $\mathbf{Y} \in \mathbb{R}^{N \times C}$ denotes the $C$-classes of node labels. Then, training a GNN model on $\mathcal{S}$ for node classification objective can be denoted as:

$$\min_{\boldsymbol{\theta}_{\mathcal{S}}} \mathcal{L}_{\mathrm{cls}} \left( \hat{\mathbf{Y}}, \mathbf{Y} \right), \text{ where } \mathbf{Z}_{\mathcal{S}}, \hat{\mathbf{Y}} = \mathrm{GNN}_{\boldsymbol{\theta}_{\mathcal{S}}}(\mathbf{X}, \mathbf{A}). \tag{1}$$

where $\boldsymbol{\theta}_{\mathcal{S}}$ denotes the parameters of GNN trained on $\mathcal{S}$, $\mathbf{Z}_{\mathcal{S}} \in \mathbb{R}^{N \times d_1}$ is the output node embedding of graph $\mathcal{S}$ from $\mathrm{GNN}_{\boldsymbol{\theta}_{\mathcal{S}}}$, and $\hat{\mathbf{Y}} \in \mathbb{R}^{N \times C}$ denotes GNN predicted node labels. By optimizing the node classification loss function $\mathcal{L}_{\mathrm{cls}}$ between GNN predictions $\hat{\mathbf{Y}}$ and ground-truth node labels $\mathbf{Y}$, *i.e.*, cross-entropy loss, we could obtain a well-trained GNN with optimal weight parameters $\boldsymbol{\theta}_{\mathcal{S}}^*$, denoted as $\mathrm{GNN}_{\mathcal{S}}^*$. Then, the node classification accuracy can be calculated to reflect GNN performance as: $\mathrm{Acc}(\mathcal{S}) = \sum_{i=1}^{N} (\hat{y}_i == y_i)/N$, which indicates the percentage of correctly predicted node labels between the GNN predicted labels $\hat{y}_i \in \hat{\mathbf{Y}}$ and ground truths $y_i \in \mathbf{Y}$. Given an unseen and unlabeled graph $\mathcal{T} = (\mathbf{X}', \mathbf{A}')$ including $M$ nodes with its features $\mathbf{X}' \in \mathbb{R}^{M \times d}$ and structures $\mathbf{A}' \in \mathbb{R}^{M \times M}$, we assume the covariate shift between $\mathcal{S}$ and $\mathcal{T}$, where the distribution shift mainly lies in node numbers, node context features, and graph structures, but the label space of $\mathcal{T}$ keeps the same with $\mathcal{S}$, *i.e.*, all nodes in $\mathcal{T}$ are constrained in the same $C$-classes. Due to the absence of ground-truth node labels, we could NOT directly calculate the node classification accuracy to assess the performance of the well-trained $\mathrm{GNN}_{\mathcal{S}}^*$ on $\mathcal{T}$. In light of this, we present a new research problem for GNNs as:

**Definition of GNN Model Evaluation.** Given the observed training graph $\mathcal{S}$, its well-trained model $\mathrm{GNN}_{\mathcal{S}}^*$, and an unlabeled unseen graph $\mathcal{T}$ as inputs, the **goal** of GNN model evaluation aims to learn an accuracy estimation model $f_{\boldsymbol{\phi}}(\cdot)$ parameterized by $\phi$ as:

$$\mathrm{Acc}(\mathcal{T}) = f_{\boldsymbol{\phi}}(\mathrm{GNN}_{\mathcal{S}}^*, \mathcal{T}), \tag{2}$$

where $f_{\boldsymbol{\phi}} : (\mathrm{GNN}_{\mathcal{S}}^*, \mathcal{T}) \to a$ and $a \in \mathbb{R}$ is a scalar denoting the overall node classification accuracy $\mathrm{Acc}(\mathcal{T})$ for all unlabeled nodes of $\mathcal{T}$. When the context is clear, we will use $f_{\boldsymbol{\phi}}(\mathcal{T})$ for simplification.

## 3   The Proposed Method

### 3.1   Overall Framework of GNN Model Evaluation

As shown in Fig. 2, the proposed overall framework of GNN model evaluation contains two stages: (1) DiscGraph set construction; and (2) `GNNEvaluator` training and inference.

**Stage 1.** Given the observed training graph $\mathcal{S}$, we first extract a seed graph from it followed by augmentations, leading to a set of meta-graphs from it for simulating any potential unseen graph distributions in practice. Then, the meta-graph set $\mathcal{G}_{\mathrm{meta}}$ and the observed training graph $\mathcal{S}$ are fed into the well-trained GNN for obtaining latent node embeddings, *i.e.*, $\mathbf{Z}_{\mathrm{meta}}^{(i,*)}$ and $\mathbf{Z}_{\mathcal{S}}^*$, respectively. After, a

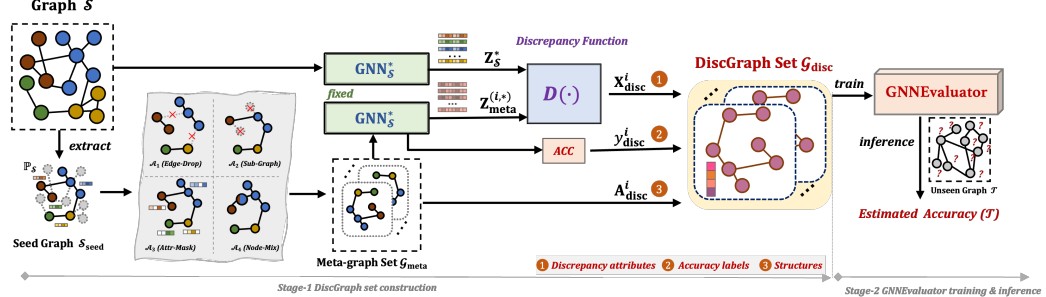

Figure 2: Overall two-stage pipeline of GNN model evaluation with (1) DiscGraph set construction and (2) `GNNEvaluator` training and inference.

discrepancy measurement function $D(\cdot)$ works on these two latent node embeddings, leading to ① discrepancy node attributes $\mathbf{X}_{\text{disc}}^i$. Meanwhile, the output node class predictions of well-trained GNN on each meta-graph are used to calculate the node classification accuracy according to its ground-truth node labels, leading to the new ② scalar accuracy labels $y_{\text{disc}}^i$. Along with ③ graph structures $\mathbf{A}_{\text{disc}}^i$ from each meta-graph, these three important components composite the set of discrepancy meta-graphs (DiscGraph) in expressive graph structural discrepancy representations.

**Stage 2.** In the second stage, the representative DiscGraph set is taken as effective supervision to train a `GNNEvaluator` through accuracy regression. During the inference, the trained `GNNEvaluator` could directly output the node classification accuracy of the in-service GNN on the unseen test graph $\mathcal{T}$ without any node class labels.

### 3.2 Discrepancy Meta-graph Set Construction

The critical challenge in developing GNN model evaluation methods is complex and diverse graph data distribution discrepancies between various real-world unseen test graphs and the observed training graph. As the in-service GNN in practice fits the observed training graph well, making inferences on various and diverse unseen test graphs would incur significant performance uncertainty for GNN model evaluation, especially when the node classification accuracy can not be calculated due to label absence.

In light of this, we propose to construct a set of diverse graphs to simulate wide-range discrepancies of potential unseen graph distributions. Specifically, such a graph set should have the following characteristics: (1) *sufficient quantity:* it should contain a relatively sufficient number of graphs with diverse node context and graph structure distributions; (2) *represented discrepancy:* the node attributes of each graph should indicate its distribution distance gap towards the observed training graph; (3) *known accuracy:* each graph should be annotated by node classification accuracy as its label.

To address the characteristic of (1) *sufficient quantity*, we introduce a meta-graph simulation strategy to synthesize a wide variety of meta-graphs for representing any potential unseen graph distributions that may be encountered in practice. Specifically, we extract a seed sub-graph $\mathcal{S}_{\text{seed}}$ from the observed training graph $\mathcal{S}$. The principle of seed sub-graph selection strategy is that $\mathcal{S}_{\text{seed}}$ involves the least possible distribution shift within $\mathbb{P}_{\mathcal{S}}$ from the observed training graph, and shares the same label space with $\mathcal{S}$, satisfying the assumption of covariate shift. Then, $\mathcal{S}_{\text{seed}}$ is fed to a pool of graph augmentation operators as

$$\mathcal{A} = \{\mathcal{A}_1(\text{EDGEDROP}), \mathcal{A}_2(\text{SUBGRAPH}), \mathcal{A}_3(\text{ATTRMASK}), \mathcal{A}_4(\text{NODEMIX})\}, \quad (3)$$

which belongs to the distribution $\mathbb{P}_{\mathcal{A}}(p_i, |\epsilon)$ with $\{p_i\}_{i=1}^4 \in (0,1)$ represent the augmentation ratio on $\mathcal{S}_{\text{seed}}$ for each augmentation type, and $\epsilon$ determines the probability of sampling a particular augmentation type. For instance, $p_1 = 0.5$ means dropping 50% edges in $\mathcal{S}_{\text{seed}}$. In this way, we can obtain a set of meta-graphs $\mathcal{G}_{\text{meta}} = \{g_{\text{meta}}^i\}_{i=1}^K$ with $K$ numbers of graphs, and we have $\mathcal{G}_{\text{meta}} \sim \mathbb{P}_{\mathcal{A}} \times \mathbb{P}_{\mathcal{S}} : \mathcal{S}_{\text{seed}} \rightarrow g_{\text{meta}}^i$. For each meta-graph $g_{\text{meta}}^i = \{\mathbf{X}_{\text{meta}}^i, \mathbf{A}_{\text{meta}}^i, \mathbf{Y}_{\text{meta}}^i\}$, where

$\mathbf{X}_{\text{meta}}^i \in \mathbb{R}^{M_i \times d}$, $\mathbf{A}_{\text{meta}}^i \in \mathbb{R}^{M_i \times M_i}$, and $\mathbf{Y}_{\text{meta}}^i \in \mathbb{R}^{M_i \times C}$ have $M_i$ number of nodes, $d$-dimensional features, and known node labels from $\mathcal{S}_{\text{seed}}$ belonging to $C$ classes.

To incorporate the characteristic of (2) *represented discrepancy*, we fully exploit the outputs of latent node embeddings and node class predictions from well-trained $\text{GNN}_{\mathcal{S}}^*$, and integrate various training-test graph distribution differences into discriminative discrepancy representations. Specifically, given the $\text{GNN}_{\mathcal{S}}^*$ learned latent node embeddings $\mathbf{Z}_{\mathcal{S}}^* = \text{GNN}_{\mathcal{S}}^*(\mathbf{X}, \mathbf{A})$, and $\mathbf{Z}_{\text{meta}}^{(i,*)}, \hat{\mathbf{Y}}_{\text{meta}}^{(i,*)} = \text{GNN}_{\mathcal{S}}^*(\mathbf{X}_{\text{meta}}^i, \mathbf{A}_{\text{meta}}^i)$, where $\mathbf{Z}_{\mathcal{S}}^* \in \mathbb{R}^{N \times d_1}$ and $\mathbf{Z}_{\text{meta}}^{(i,*)} \in \mathbb{R}^{M_i \times d_1}$, we derive a distribution discrepancy measurement function $D(\cdot)$ to calculate the discrepancy meta-graph node attributes as:

$$\mathbf{X}_{\text{disc}}^i = D(\mathbf{Z}_{\text{meta}}^{(i,*)}, \mathbf{Z}_{\mathcal{S}}^*) = \frac{\mathbf{Z}_{\text{meta}}^{(i,*)} \mathbf{Z}_{\mathcal{S}}^{* \text{T}}}{\left\| \mathbf{Z}_{\text{meta}}^{(i,*)} \right\|_2 \cdot \| \mathbf{Z}_{\mathcal{S}}^* \|_2}, \tag{4}$$

where $\mathbf{X}_{\text{disc}}^i \in \mathbb{R}^{M_i \times N}$ reflects the node-level distribution discrepancy between each $g_{\text{meta}}^i$ and $\mathcal{S}$ with the well-trained $\text{GNN}_{\mathcal{S}}^*$. Each element in $x_{u,v}^i \in \mathbf{X}_{\text{disc}}^i$ denotes the node embedding discrepancy gap between a certain node $u$ in the $i$-th meta-graph and a node $v$ in the observed training graph. Taking this representation as node attributes could effectively integrate representative node-level discrepancy produced by well-trained GNN's node embedding space.

Meanwhile, the characteristic of (3) *known accuracy* can be involved with the outputs of node class predictions produced by $\text{GNN}_{\mathcal{S}}^*$ on meta-graph $\hat{\mathbf{Y}}_{\text{meta}}^{(i,*)} = \{\hat{y}_{\text{meta}(i,*)}^j\}_{j=1}^{M_i}$. We calculate the node classification accuracy on the meta-graph given its ground-truth node class labels $\mathbf{Y}_{\text{meta}}^i = \{y_{\text{meta}(i)}^j\}_{j=1}^{M_i}$ as:

$$y_{\text{disc}}^i = \text{Acc}(g_{\text{meta}}^i) = \frac{\sum_{j=1}^{M_i} (\hat{y}_{\text{meta}(i,*)}^j == y_{\text{meta}(i)}^j)}{M_i}, \tag{5}$$

where $y_{\text{meta}(i)}^j$ and $\hat{y}_{\text{meta}(i,*)}^j$ denote the ground truth label and $\text{GNN}_{\mathcal{S}}^*$ predicted label of $j$-th node in the $i$-th graph of the meta-graph set, respectively. Note that $y_{\text{disc}}^i \in \mathbb{R}$ is a continuous scalar denoting node classification accuracy under specific $\text{GNN}_{\mathcal{S}}^*$ within the range of $(0, 1)$.

In this way, by incorporating all these characteristics with the discrepancy node attributes ($\mathbf{X}_{\text{disc}}^i$, the scalar node classification accuracy label $y_{\text{disc}}^i$, and the graph structure from meta-graph as $\mathbf{A}_{\text{disc}}^i = \mathbf{A}_{\text{meta}}^i$ for indicating discrepancy node interconnections, we could derive the final discrepancy meta-graph set, *i.e.*, DiscGraph set, as

$$\mathcal{G}_{\text{disc}} = \left\{ g_{\text{disc}}^i \right\}_{i=1}^K, \text{where } g_{\text{disc}}^i = (\mathbf{X}_{\text{disc}}^i, \mathbf{A}_{\text{disc}}^i, y_{\text{disc}}^i). \tag{6}$$

The proposed DiscGraph set $\mathcal{G}_{\text{dist}}$ contains a sufficient number of graphs with diverse node context and graph structure distributions, where each discrepancy meta-graph contains the latent node embedding based discrepancy node attributes and the node class prediction based accuracy label, according to the well-trained GNN's outputs, along with diverse graph structures. All these make the proposed DiscGraph set a discriminative graph structural discrepancy representation for capturing wide-range graph data distribution discrepancies.

### 3.3 GNNEvaluator Training and Inference

**Training.** Given our constructed DiscGraph set $\mathcal{G}_{\text{disc}} = \left\{ g_{\text{disc}}^i \right\}_{i=1}^K$ and $g_{\text{disc}}^i = (\mathbf{X}_{\text{disc}}^i, \mathbf{A}_{\text{disc}}^i, y_{\text{disc}}^i)$ in Eq. (6), we use it to train a GNN regressor with $f_\phi : g_{\text{disc}}^i \rightarrow a$ for evaluating well-trained GNNs, which we name as GNNEvaluator. Specifically, the proposed GNNEvaluator takes a two-layer GCN architecture as the backbone, followed by a pooling layer to average the representation of all nodes of each $g_{\text{disc}}^i$, and then maps the averaged embedding to a scalar node classification accuracy on the whole graph. The objective function of our GNN regressor can be written as:

$$\min_\phi \sum_{i=1}^K \mathcal{L}_{\text{reg}}(f_\phi(\mathbf{X}_{\text{disc}}^i, \mathbf{A}_{\text{disc}}^i), y_{\text{disc}}^i), \tag{7}$$

where $\mathcal{L}_{\text{reg}}$ is the regression loss, *i.e.*, the mean square error (MSE) loss.

**Inference.** During the inference in the practical GNN model evaluation, we have: (1) to-be-evaluated $\text{GNN}_{\mathcal{S}}^*$, and (2) the unseen test graph $\mathcal{T} = (\mathbf{X}', \mathbf{A}')$ without labels. The first thing is to calculate

the discrepancy node attributes on the unseen test graph $\mathcal{T}$ towards the observed training graph $\mathcal{S}$ according to Eq. (4), so that we could obtain $\mathbf{X}_{\text{disc}}^{\mathcal{T}} = D(\mathbf{Z}_{\mathcal{T}}^*, \mathbf{Z}_{\mathcal{S}}^*)$, where $\mathbf{Z}_{\mathcal{T}}^* = \text{GNN}_{\mathcal{S}}^*(\mathbf{X}', \mathbf{A}')$. Along with the unseen graph structure $\mathbf{A}_{\text{disc}}^{\mathcal{T}} = \mathbf{A}'$, the proposed `GNNEvaluator` could directly output the node classification accuracy of $\text{GNN}_{\mathcal{S}}^*$ on $\mathcal{T}$ as:

$$\text{Acc}(\mathcal{T}) = \hat{y}_{\text{dist}}^{\mathcal{T}} = f_{\phi^*}(\mathbf{X}_{\text{disc}}^{\mathcal{T}}, \mathbf{A}_{\text{disc}}^{\mathcal{T}}), \tag{8}$$

where $\phi^*$ denotes the optimal `GNNEvaluator` weight parameters trained by our constructed Disc-Graph set $\mathcal{G}_{\text{dist}}$.

## 4  Experiments

This section empirically evaluates the proposed `GNNEvaluator` on real-world graph datasets for node classification task. In all experiments, to-be-evaluated GNN models have been well-trained on observed training graphs and keep FIXED in GNN model evaluation. We only vary the unseen test graph datasets and evaluate various different types and architectures of well-trained GNNs. Throughout our proposed entire two-stage GNN model evaluation framework, we are unable to access the labels of unseen test graphs. We only utilize these labels for the purpose of obtaining true error estimates for experimental demonstration.

We investigate the following questions to verify the effectiveness of the proposed method. **Q1:** How does the proposed `GNNEvaluator` perform in evaluating well-trained GNNs' node classification accuracy (Sec. 4.2)? **Q2:** How does the proposed `GNNEvaluator` perform when conducting an ablation study regarding the DiscGraph set component? (Sec. 4.3) **Q3:** What characteristics does our constructed DiscGraph set have (Sec. 4.4)? **Q4:** How many DiscGraphs in the set are needed for the proposed `GNNEvaluator` to perform well (Sec. 4.5)?

### 4.1  Experimental Settings

**Datasets.** We perform experiments on three real-world graph datasets, *i.e.*, DBLPv8, ACMv9, and Citationv2, which are citation networks from different original sources (DBLP, ACM, and Microsoft Academic Graph, respectively) by following the processing of [29]. Each dataset shares the same label space with six categories of node class labels, including 'Database', 'Data mining', 'Artificial intelligent', 'Computer vision', 'Information Security', and 'High-performance computing'. We evaluate our proposed `GNNEvaluator` with the following six cases, *i.e.*, A→D, A→C, C→A, C→D, D→A, and D→C, where A, C, and D denote ACMv9, DBLPv8, and Citationv2, respectively. Each arrow denotes the estimation of `GNNEvaluator` trained by the former observed graph and tested on the latter unseen graph. Note that, in all stages of `GNNEvaluator` training and inference, we do not access the labels of the latter unseen graph. More detailed statistical information of the used dataset can be found in Appendix B.

**GNN Models and Evaluation.** We evaluate four commonly-used GNN models, including GCN [17], GraphSAGE [12] (*abbr.* SAGE), GAT [28], and GIN [31], as well as the baseline MLP model that is prevalent for graph learning. For each model, we train it on the training set of the observed graph under the transductive setting, until the model achieves the best node classification on its validation set following the standard training process, *i.e.*, the 'well-trained' GNN model. To minimize experimental variability, we train each type of GNN with five random seeds. That means, for instance, we have five well-trained GCN models on the same fully-observed graph with the same hyper-parameter space but only five different random seeds. More details of these well-trained GNN models are provided in Appendix D. We report the Mean Absolute Error (MAE), the average absolute difference between the ground truth accuracy (in percentage) on the unseen graph and the estimated accuracy from ours and baselines on the same unseen graph, across different random seeds for each evaluated GNN. The smaller MAE denotes better GNN model evaluation performance.

**Baseline Methods.** As our method is the first GNN model evaluation approach, there is no available baseline for direct comparisons. Therefore, we evaluate our proposed approach by comparing it to three convolutional neural network (CNN) model evaluation methods applied to image data. Note that existing CNN model evaluation methods are hard to be directly applied to GNNs, since GNNs have entirely different convolutional architectures working on different data types. Compared with Euclidean image data, graph structural data distributions have more complex and wider variations due to their non-independent node-edge interconnections. Therefore, we make necessary adaptations to

Table 1: Mean Absolute Error (MAE) performance of different GNN models across different random seeds. (GNNs are well-trained on the ACMv9 dataset and evaluated on the unseen and unlabeled Citationv2 and DBLPv8 datasets, *i.e.*, A→C and A→D, respectively. Best results are in bold.)

| Methods | ACMv9→Citationv2 | | | | | | ACMv9→DBLPv8 | | | | | |
|---|---|---|---|---|---|---|---|---|---|---|---|---|
| | GCN | SAGE | GAT | GIN | MLP | *Avg.* | GCN | SAGE | GAT | GIN | MLP | *Avg.* |
| ATC-MC [9] | 4.49 | 8.40 | 4.37 | 18.40 | 34.33 | 14.00 | 21.96 | 24.20 | 30.30 | 24.06 | 26.62 | 25.43 |
| ATC-MC-c [9] | **2.41** | 5.74 | 4.67 | 22.00 | 51.41 | 17.25 | 31.15 | 30.55 | 30.18 | 29.71 | 45.81 | 33.48 |
| ATC-NE [9] | 3.97 | 8.02 | 4.28 | 17.35 | 38.87 | 14.50 | 22.93 | 24.78 | 30.50 | 23.74 | 31.13 | 26.62 |
| ATC-NE-c [9] | 4.44 | 6.09 | **3.30** | 23.95 | 44.62 | 16.48 | 34.42 | 28.31 | 27.02 | 30.28 | 39.28 | 31.86 |
| Thres. ($\tau = 0.7$) [6] | 32.64 | 35.81 | 33.63 | 50.76 | 35.28 | 37.63 | 9.59 | 12.14 | 14.30 | 32.67 | 39.72 | 21.68 |
| Thres. ($\tau = 0.8$) [6] | 26.30 | 29.60 | 26.18 | 49.25 | 35.87 | 33.44 | **2.63** | 7.44 | 14.47 | 32.20 | 40.31 | 19.41 |
| Thres. ($\tau = 0.9$) [6] | 17.56 | 21.34 | 16.38 | 46.53 | 36.08 | 27.58 | 8.20 | 7.42 | 16.07 | 31.47 | 40.56 | 20.74 |
| AutoEval-G [6] | 18.94 | 26.19 | 26.12 | 50.86 | 32.40 | 30.90 | 2.77 | **2.54** | 7.25 | 48.68 | 29.95 | 18.24 |
| GNNEvaluator (**Ours**) | 4.85 | **4.11** | 12.23 | **10.14** | **22.20** | **10.71** | 11.80 | 14.88 | **6.36** | **13.78** | **17.49** | **12.86** |

Table 2: Mean Absolute Error (MAE) performance of different GNN models across different random seeds. (GNNs are well-trained on the Citationv2 dataset and evaluated on the unseen and unlabeled ACMv9 and DBLPv8 datasets, *i.e.*, C→A and C→D, respectively. Best results are in bold.)

| Methods | Citationv2→ACMv9 | | | | | | Citationv2→DBLPv8 | | | | | |
|---|---|---|---|---|---|---|---|---|---|---|---|---|
| | GCN | SAGE | GAT | GIN | MLP | *Avg.* | GCN | SAGE | GAT | GIN | MLP | *Avg.* |
| ATC-MC [9] | 9.50 | 13.40 | 8.28 | 35.51 | 43.40 | 22.02 | 22.57 | **1.37** | 21.87 | 29.24 | 35.20 | 22.05 |
| ATC-MC-c [9] | 6.93 | 11.75 | **6.70** | 38.93 | 57.43 | 24.35 | 33.67 | 4.92 | 28.23 | 30.89 | 52.59 | 30.06 |
| ATC-NE [9] | 8.86 | 13.04 | 7.87 | 34.88 | 47.49 | 22.42 | 23.97 | 1.86 | 23.74 | 28.96 | 39.72 | 23.65 |
| ATC-NE-C [9] | 7.73 | 13.94 | 7.63 | 41.17 | 62.96 | 26.69 | 37.16 | 4.66 | 29.43 | 31.66 | 58.95 | 32.37 |
| Thres. ($\tau = 0.7$) [6] | 37.33 | 36.61 | 31.68 | 58.91 | 34.33 | 39.77 | 10.70 | 23.05 | 12.74 | 34.60 | 38.29 | 23.88 |
| Thres. ($\tau = 0.8$) [6] | 29.62 | 28.95 | 22.77 | 57.48 | 34.53 | 34.67 | 5.65 | 15.01 | 7.61 | 34.36 | 38.43 | 20.21 |
| Thres. ($\tau = 0.9$) [6] | 19.59 | 19.06 | 11.37 | 55.72 | 34.56 | 28.06 | 10.65 | 8.28 | 8.07 | 34.00 | 38.44 | 19.89 |
| AutoEval-G [6] | 23.01 | 31.24 | 26.74 | 59.66 | 35.02 | 28.28 | **2.57** | 16.52 | 6.96 | 19.20 | 32.24 | 24.59 |
| GNNEvaluator (**Ours**) | **5.45** | **8.53** | 9.61 | **29.77** | **28.52** | **16.38** | 11.64 | 7.02 | **5.58** | **6.46** | 22.87 | **10.71** |

enable these methods to work on graph-structured data for GNN model evaluation. Specifically, we compare: (1) *Average Thresholded Confidence (ATC) score* [9], which learns a threshold on CNN's confidence to estimate the accuracy as the fraction of unlabeled images whose confidence scores exceed the threshold. We consider its four variants, denoting as ATC-NE, ATC-NE-c, ATC-MC, and ATC-MC-c, where ATC-NE and ATC-MC calculate negative entropy and maximum confidence scores, respectively, and '-c' denotes their confidence calibrated versions. In our adaptation, we use the training set of the observed graph under the transductive setting to calculate these confidence scores, and use the validation set for their calibrated versions. (2) *Threshold-based Method*, which is introduced by [6] and determines three threshold values for $\tau = \{0.7, 0.8, 0.9\}$ on the output softmax logits of CNNs and calculates the percentage of images in the entire dataset whose maximum entry of logits are greater than the threshold $\tau$, which indicates these images are correctly classified. We make an adaptation by using the GNN softmax logits and calculating the percentage of nodes in a single graph. (3) *AutoEval Method* [6], which conducts the linear regression to learn the CNN classifier's accuracy based on the domain gap features between the observed training image data and unseen real-world unlabeled image data. We adapt it to GNN models on graph structural data by calculating the Maximum Mean Discrepancy (MMD) measurement as the graph domain gap features extracted from our meta-graph set, followed by the linear regression, which we name as AutoEval-G variant. More details of baseline methods can be found in the Appendix C.

## 4.2 GNN Model Evaluation Results

In Table 1, Table 2, and Table 3, we report MAE results of evaluating different GNN models across different random seeds. In general, we observe that our proposed `GNNEvaluator` significantly outperforms the other model evaluation baseline methods in all six cases, achieving the lowest average MAE scores of all GNN model types, *i.e.*, 10.71 and 12.86 in A→C and A→D cases, 16.38 and 10.71 in C→A and C→D cases, and 6.79 and 7.80 in D→A and D→C cases, respectively.

More specifically, we observe that some baseline methods achieve the lowest MAE for a certain GNN model type on a specific case. Taking GCN as an example in Table 1, ATC-MC-c performs best with 2.41 MAE under A→C case. However, it correspondingly has 31.15 worst MAE under A→D case. That means, for a GCN model that is well-trained by ACMv9 dataset, ATC-MC-c provides model evaluation results with significant performance variance under different unseen graphs. Such high-

Table 3: Mean Absolute Error (MAE) performance of different GNN models across different random seeds. (GNNs are well-trained on the DBLPv8 dataset and evaluated on the unseen and unlabeled ACMv9 and Citationv2 datasets, *i.e.*, D→A and D→C, respectively. Best results are in bold.)

| Methods | DBLPv8→ACMv9 | | | | | | DBLPv8→Citationv2 | | | | | |
|---|---|---|---|---|---|---|---|---|---|---|---|---|
| | GCN | SAGE | GAT | GIN | MLP | *Avg.* | GCN | SAGE | GAT | GIN | MLP | *Avg.* |
| ATC-MC [9] | 4.98 | 31.49 | 37.08 | 28.74 | 30.42 | 26.54 | 5.47 | 29.72 | 37.43 | 32.07 | 36.42 | 28.22 |
| ATC-MC-C [9] | 3.42 | 32.83 | 40.31 | 37.68 | 34.72 | 29.79 | **4.66** | 28.67 | 43.89 | 38.13 | 41.64 | 31.40 |
| ATC-NE [9] | 2.83 | 27.67 | 40.90 | 35.88 | 36.03 | 28.66 | 4.25 | 26.04 | 39.38 | 38.38 | 45.15 | 30.64 |
| ATC-NE-C [9] | 12.95 | 20.86 | 39.34 | 43.41 | 38.85 | 31.08 | 16.87 | 15.77 | 40.26 | 47.64 | 47.61 | 33.63 |
| Thres. ($\tau = 0.7$) [6] | 21.66 | 32.69 | 39.19 | 30.10 | 27.80 | 30.29 | 27.22 | 35.48 | 34.35 | 37.99 | 30.36 | 33.08 |
| Thres. ($\tau = 0.8$) [6] | 18.63 | 29.34 | 37.25 | 25.80 | 20.34 | 26.27 | 22.70 | 30.86 | 32.48 | 32.73 | 22.36 | 28.23 |
| Thres. ($\tau = 0.9$) [6] | 16.12 | 23.81 | 43.26 | 23.73 | 12.98 | 23.98 | 15.63 | 26.07 | 35.88 | 27.15 | 12.20 | 23.39 |
| AutoEval-G [6] | 7.28 | **9.72** | 12.05 | 14.17 | 22.07 | 13.06 | 21.13 | 15.11 | 5.65 | 12.33 | 31.90 | 17.22 |
| GNNEvaluator (**Ours**) | **2.46** | 10.27 | **6.94** | **8.86** | **5.42** | 6.79 | 11.68 | **7.83** | **3.97** | **9.62** | **5.92** | 7.80 |

variance evaluation performance would significantly limit the practical applications of ATC-MC-c, as it only performs well on certain unseen graphs, incurring more uncertainty in GNN model evaluation. Similar limitations can also be observed in Table 2 for ATC-MC-c on evaluating GAT model, with 6.70 lowest MAE in C→A but 28.23 high MAE in C→D. Moreover, in terms of AutoEval-G method on GraphSAGE model evaluation in Table 3, it achieves the lowest MAE of 9.72 in D→A case, but a high MAE of 15.11 in C→A case. In contrast, our proposed GNNEvaluator has smaller MAE variance for different unseen graphs under the same well-trained models. For instance, our GNNEvaluator achieves 4.85 MAE on GCN under A→C case, and 11.80 MAE on GCN in A→D case, which is better than ATC-MC-c with 2.41 and 31.15 on the same cases. All these results verify the effectiveness and consistently good performance of our proposed method on diverse unseen graph distributions for evaluating different GNN models.

## 4.3 Ablation Study

We conduct an ablation study to verify the effectiveness of the proposed DiscGraph set component for our proposed GNN model evaluation pipeline. Concretely, we replace the DiscGraph set with the meta-graph set for GNNEvaluator training, which does not involve discrepancy node attributes calculated based on specific well-trained GNNs' node embedding space with the discrepancy measurement function. The output node classification accuracy (%) of the proposed GNNEvaluator for evaluating five GAT models under different seeds are shown in Fig. 3 with lines for the proposed DiscGraph set (w/ DiscAttr) in green and the meta-graph set (w/o DiscAttr) in blue, respectively. And we also list the ground-truth node classification accuracy in histograms for comparison.

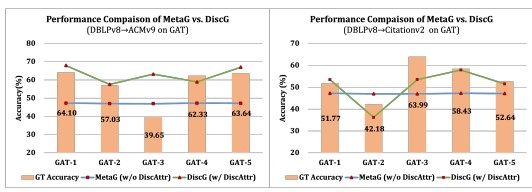

Figure 3: Ablation study of the proposed DiscGraph set with discrepancy node attributes (w/ DiscAttr) in green lines *vs.* the meta-graph set without discrepancy node attributes (w/o DiscAttr) in blue lines. And the ground-truth node classification accuracy is in histograms for reference.

Generally, we observe that the proposed DiscGraph set (w/ DiscAttr) trained GNNEvaluator has better performance for GAT model evaluation than that trained by the meta-graph set (w/o DiscAttr), as all green lines are closer to the ground-truth histograms than the blue line, reflecting the effectiveness of the proposed the proposed DiscGraph set, especially the discrepancy node attributes with the discrepancy measurement function. Moreover, the meta-graph set (w/o DiscAttr) trained GNNEvaluator almost produces the same accuracy on a certain GNN type, *i.e.*, its performance stays the same on all five GATs, making it fail to evaluate a certain GNN model type under different optimal model parameters. That is because, compared with the proposed DiscGraph set, the meta-graph set lacks the exploration of the latent node embedding space of well-trained GNNs and only utilizes the output node class predictions from the well-trained GNNs. In contrast, the proposed DiscGraph set fully exploits both latent node embedding space and output node class predictions of well-trained GNNs for comprehensively modeling the discrepancy between the observed training graph and unseen test graph, enabling its superior ability for GNN model evaluation.

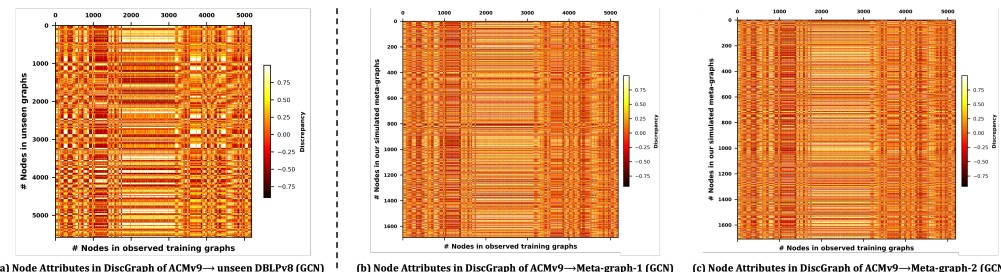

(a) Node Attributes in DiscGraph of ACMv9→ unseen DBLPv8 (GCN)  (b) Node Attributes in DiscGraph of ACMv9→Meta-graph-1 (GCN)  (c) Node Attributes in DiscGraph of ACMv9→Meta-graph-2 (GCN)

Figure 4: Visualizations on discrepancy node attributes in the proposed DiscGraph set for GCN model evaluation. Darker color denotes a larger discrepancy.

## 4.4 Analysis of Discrepancy Meta-graph Set

We visualize the discrepancy node attributes in the proposed DiscGraph set on GCN model evaluation trained on ACMv9 and tested on unseen DBLPv8 in Fig. 4 (a). Taking this as a reference, we present visualizations of discrepancy node attributes derived between ACMv9 and arbitrary two meta-graphs from our created meta-graph set in Fig. 4 (c) and (d). Darker color close to black indicates a larger discrepancy in the output node embedding space of a well-trained GNN, and lighter color close to white indicates a smaller discrepancy. As can be observed, the discrepancy pairs of (ACMv9, unseen DBLPv8), (ACMv9, Meta-graph-1), and (ACMv9, Meta-graph-2) generally show similar heat map patterns in terms of discrepancy. This indicates the effectiveness of the proposed meta-graph set simulation strategy, which could capture potential unseen graph data distributions by synthesizing diverse graph data with various graph augmentations. More detailed statistical information about the proposed DiscGraph set, including the number of graphs, the average number of nodes and edges, and the accuracy label distributions are provided in Appendix E.

## 4.5 Hyper-parameter Analysis of GNNEvaluator

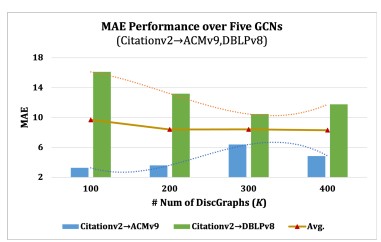

Figure 5: Effects of different numbers of DiscGraphs ($K$) for GNNEvaluator training. Histograms denote MAE for $C \rightarrow A$ and $C \rightarrow D$ cases over five GCNs under different $K$, and the line indicates the average MAE of such two evaluation cases.

We test the effects of different numbers of discrepancy meta-graphs ($K$) for GNNEvaluator training in Fig. 5. Concretely, we utilize the number of DiscGraphs for $K = [100, 200, 300, 400]$ to train the proposed GNNEvaluator for evaluating five GCN models that are well-trained on Citationv2 dataset and make inferences on unseen ACMv9 and DBLPv8 datasets. Considering the average performance under two transfer cases reflected in the line, despite $K = 100$ having a higher MAE, the average MAE over two cases does not change much with the number $K$ for training the proposed GNNEvaluator. This indicates that the proposed GNNEvaluator can effectively learn to accurately predict node attributes by training on an appropriate number of DiscGraphs, without the need for a significantly excessive amount of training examples.

## 5 Conclusion

In this work, we have studied a new problem, GNN model evaluation, for understanding and evaluating the performance of well-trained GNNs on unseen and unlabeled graphs. A two-stage approach is proposed, which first generates a diverse meta-graph set to simulate and capture the discrepancies of different graph distributions, based on which a GNNEvaluator is trained to predict the accuracy of a well-trained GNN model on unseen graphs. Extensive experiments with real-world unseen and unlabeled graphs could verify the effectiveness of our proposed method for GNN model evaluation. Our method assumes that the class label space is unchanged across training and testing graphs though covariate shifts may exist between the two. We will look into relaxing this assumption and address a broader range of natural real-world graph data distribution shifts in the future.

## Acknowledgment

In this work, S. Pan was supported by an Australian Research Council (ARC) Future Fellowship (FT210100097), M. Zhang was supported by National Natural Science Foundation of China (NSFC) grant (62306084), and C. Zhou was supported by the CAS Project for Young Scientists in Basic Research (YSBR-008).

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

# Appendix

This is the appendix of our work: **'GNNEvaluator: Evaluating GNN Performance On Unseen Graphs Without Labels'**. In this appendix, we present additional information regarding the *GNN model evaluation* problem and corresponding solutions, including in-depth discussions that distinguish our method from related works, dataset statistics used in our experiments, comprehensive details about our compared baseline methods, training details with the performance of well-trained GNN models on both the test set of the observed graph and the unseen test graphs (for ground-truth reference), and more experimental results with detailed analysis.

# A  Related Work

**Predicting Model Generalization Error.** Our work is relevant to the line of research on predicting model generalization error, which aims to develop a good estimator of a model's classifier performance on unlabeled data from the unknown distributions in the target domain, when the estimated models' classifier has been trained well in the source domain with labeled data [6, 9, 34, 5, 10, 4]. Typically, Guillory et al. [10] proposed to estimate a classifier's performance of convolutional neural network (CNN) models on image data based on a designed criterion, named difference of confidences (DoC), that estimates and reflects model accuracy. And Garg et al. [9] proposed to learn a score-based Average Thresholded Confidence (ATC) by leveraging the softmax probability of a CNN classifier, whose accuracy is estimated as the fraction of unlabeled images that receive a score above that threshold. In contrast, Deng et al. [6, 4] directly predicted CNN classifier accuracy by deriving distribution distance features between training and test images with a linear regression model.

However, these existing methods mostly focus on evaluating CNN model classifiers on image data in computer vision, and the formulation of evaluating GNNs for graph structural data still remains under-explored in graph machine learning. Concretely, conducting the model evaluation on GNNs for graph structural data has two critical challenges: (1) different from Euclidean image data, graph structural data lies in non-Euclidean space with complex and non-independent node-edge interconnections, so that its node contexts and structural characteristics significantly vary under wide-range graph data distributions, posing severer challenges for GNN model evaluation when serving on unknown graph data distributions. (2) GNNs have entirely different convolutional architectures from those of CNNs, when GNN convolution aggregates neighbor node messages along graph structures. Such that GNNs trained on the observed training graph might well fit its graph structure, and due to the complexity and diversity of graph structures, serving well-trained GNNs on unlabeled test distributions that they have not encountered before would incur more performance uncertainty.

Hence, in this work, we first investigate the GNN model evaluation problem for graph structural data with a clear problem definition and a feasible solution, taking a significant step towards understanding and evaluating GNN models for practical GNN deployment and serving. Our proposed two-stage GNN model evaluation framework directly estimates the node classification accuracy results of specific well-trained GNNs on practical unseen graphs without labels. Our method could not only facilitate model designers to better evaluate well-trained GNNs' performance in practical serving, but also provide users with confidence scores or model selection guidance, when using well-trained GNNs for inferring on their own test graphs.

**Unsupervised Graph Domain Adaption.**  Our work is also relevant to the unsupervised graph domain adaption (UGDA) problem, whose goal is to develop a GNN model with both labeled source graphs and unlabeled target graphs for better generalization ability on target graphs. Typically, existing UGDA methods focus on mitigating the domain gap by aligning the source graph distribution with the target one. For instance, Yang et al. [33] and Shen et al. [27] optimized domain distance loss based on the statistical information of datasets, *e.g.*, maximum mean discrepancy (MMD) metric. Moreover, DANE [39] and UDAGCN [29] introduced domain adversarial methods to learn domain-invariant embeddings across the source domain and the target domain. Therefore, the critical distinction between our work and UGDA is that our work is primarily concerned with evaluating the GNNs' performance on unseen graphs without labels, rather than improving the GNNs' generalization ability when adapting to unlabeled target graphs. Besides, different from UGDA which uses the unlabeled target graphs in their model design and training stage, the GNN model evaluation problem

explored by our work is not allowed to access the unlabeled test graphs, *i.e.*, unseen in the whole GNN model evaluation process.

**Graph Out-of-distribution (OOD) Generalization & Detection.** Out-of-distribution (OOD) generalization [18, 48] on graphs aims to develop a GNN model given several different but related source domains, so that the developed model can generalize well to unseen target domains. Li et al. [18] categorized existing graph OOD generalization methodologies into three conceptually different branches, *i.e.*, data, model, and learning strategy, based on their positions in the graph machine learning pipeline. We would like to highlight that, even ODD generalization and our proposed GNN model evaluation both pay attention to the general graph data distribution shift issue, we have different purposes: our proposed GNN model evaluation aims to evaluate well-trained GNNs' performance on unseen test graphs, while ODD generalization aims to develop a new GNN model for improve its performance or generalization capability on unseen test graphs. Moreover, OOD detection [32], aims to detect test samples drawn from a distribution that is different from the training distribution, with the definition of distribution to be well-defined according to the application in the target domain. Hence, the primary difference between OOD detection and our GNN model evaluation is, OOD detection works on detecting OOD test samples at the data level and our GNN model evaluation works on evaluating well-trained GNN's performance on ODD data at the model level.

## B   Dataset Statistics

We provide the details of dataset statistics used in our experiments in Table. A1, where these datasets meet the covariate shift assumption between any pair of test cases.

Table A1: Statistical details of the experimental real-world datasets.

| Dataset | # of Nodes | # of Edges | # of Features | # of Labels |
|---------|-----------|-----------|--------------|-------------|
| ACMv9 | 7410 | 11135 | 7537 | 6 |
| Citationv2 | 4715 | 6717 | 7537 | 6 |
| DBLPv8 | 5578 | 7341 | 7537 | 6 |

## C   Baseline Methods Details

**Average Thresholded Confidence (ATC) & Its Variants.** This metric [9] learns a threshold on CNN's confidence to estimate the accuracy as the fraction of unlabeled images whose confidence scores exceed the threshold as:

$$\text{ATC} = \frac{1}{M} \sum_{j=1}^{M} \mathbf{1} \left\{ s \left( \text{Softmax} \left( f_{\boldsymbol{\theta}^*} \left( \mathbf{x}_j' \right) \right) \right) < t \right\}, \tag{9}$$

where $f_{\boldsymbol{\theta}^*}(\cdot)$ denotes the well-trained CNN's classifier with the optimal parameter $\boldsymbol{\theta}^*$, and $s(\cdot)$ denotes the score function working on the softmax prediction of $f_{\boldsymbol{\theta}^*}(\cdot)$. When the context is clear, we will use $f(\cdot)$ for simplification. We adopted two different score functions, deriving two variants as: (1) Maximum confidence variant ATC-MC with $s(f(\mathbf{x}_j')) = \max_{k \in \mathcal{Y}} f_k(\mathbf{x}_j')$; and (2) Negative Entropy variant ATC-NE with $s(f(\mathbf{x}_j')) = \sum_k f_k(\mathbf{x}_j') \log \left( f_k(\mathbf{x}_j') \right)$, where $\mathcal{Y} = \{1, 2, \ldots, C\}$ is the label space. And for $t$ in Eq. (9), it is calculated based on the validation set of the observed training set $(\mathbf{x}_u^{\text{val}}, \mathbf{y}_u^{\text{val}}) \in \mathcal{S}_{\text{val}}$:

$$\frac{1}{N_{\text{val}}} \sum_{u=1}^{N_{\text{val}}} \mathbf{1} \left\{ s \left( \text{Softmax} \left( f_{\boldsymbol{\theta}^*} \left( \mathbf{x}_u^{\text{val}} \right) \right) \right) < t \right\} = \frac{1}{N_{\text{val}}} \sum_{u=1}^{N_{\text{val}}} \mathbf{1} \left\{ p \left( \mathbf{x}_u^{\text{val}}; \boldsymbol{\theta}^* \right) \neq \mathbf{y}_u^{\text{val}} \right\}, \tag{10}$$

where $p(\cdot)$ denotes the predicted labels of samples. For all the calibration variants with '-c' in our experiments, they conduct standard Temperature Scaling [11] with following equations:

$$p_u = \max_k \text{Softmax} \left( \mathbf{z}_u / T \right)^{(k)}, \tag{11}$$

Table A2: Hyperparameters of learning rate (lr) and weight decay (wd) in training different GNNs and MLP.

| Datasets | ACMv9 | | Citationv2 | | DBLPv8 | |
|---|---|---|---|---|---|---|
| Models | lr | wd | lr | wd | lr | wd |
| GCN | 0.01 | 1.00E-05 | 0.01 | 1.00E-05 | 0.01 | 1.00E-05 |
| SAGE | 0.005 | 1.00E-06 | 0.005 | 1.00E-06 | 0.005 | 1.00E-06 |
| GAT | 0.005 | 1.00E-06 | 0.005 | 1.00E-06 | 0.005 | 1.00E-06 |
| GIN | 0.01 | 1.00E-06 | 0.01 | 1.00E-06 | 0.01 | 1.00E-06 |
| MLP | 0.001 | 1.00E-05 | 0.001 | 1.00E-05 | 0.001 | 1.00E-05 |

where $\mathbf{z}_u$ denotes the network output logits before softmax and $T$ is the temperature scaling factor.

**Threshold-based Method.** This is an intuitive solution introduced by [6], which is not a learning-based method. It follows the basic assumption that a class prediction will likely be correct when it has a high softmax output score. Then, the threshold-based method would provide the estimated accuracy of a model as:

$$\text{ACC} = \frac{\sum_{i=1}^{M} \mathbf{1}\left\{\max\left(f_{\boldsymbol{\theta}^*}(\mathbf{x}'_j)\right) > \tau\right\}}{M}, \tag{12}$$

where $\tau$ is the pre-defined thresholds as $\tau = \{0.7, 0.8, 0.9\}$ on the output softmax logits of CNNs. This metric calculates the percentage of images in the entire dataset whose maximum entry of logits are greater than the threshold $\tau$, which indicates these images are correctly classified.

**AutoEval & Our Adaption AutoEval-G.** This is a learning-based method of estimating classifier accuracy by utilizing dataset-level feature statistics [6]. AutoEval synthesizes a meta image dataset $\mathcal{D}$ (a dataset comprised of many datasets) from the observed training dataset $\mathcal{D}_{\text{ori}}$, and conducts the linear regression to learn the CNN classifier's accuracy based on the dataset-level feature statistics between the observed training image data and unseen real-world unlabeled image data. The accuracy linear regression with the dataset statistic feature $f_{\text{linear}}$ is written as:

$$f_{\text{linear}} = \text{FD}\left(\mathcal{D}_{\text{ori}}, \mathcal{D}\right) = \|\boldsymbol{\mu}_{\text{ori}} - \boldsymbol{\mu}\|_2^2 + \text{Tr}\left(\boldsymbol{\Sigma}_{\text{ori}} + \boldsymbol{\Sigma} - 2\left(\boldsymbol{\Sigma}_{\text{ori}}\boldsymbol{\Sigma}\right)^{\frac{1}{2}}\right),$$
$$\text{ACC} = w_1 f_{\text{linear}} + w_0 \tag{13}$$

where FD is the Fréchet distance [7] to measure the domain gap with the mean feature vectors $\boldsymbol{\mu}_{\text{ori}}$ and $\boldsymbol{\mu}$, the covariance matrices $\boldsymbol{\Sigma}_{\text{ori}}$ and $\boldsymbol{\Sigma}$ of $\mathcal{D}_{\text{ori}}$ and $\mathcal{D}$, respectively. And $\text{Tr}(\cdot)$ denotes the trace of the matrix, $w_1$ and $w_0$ denote the parameters of linear regression.

We make the following necessary adaptions to expand the AutoEval method to graph structural data on GNN model evaluation, deriving **AutoEval-G** by: (1) we synthesize a meta-graph set $\mathcal{G}_{\text{meta}}$ as demonstrated in Sec. 3.2 of the main manuscript; (2) we calculate the Maximum Mean Discrepancy (MMD) distance as AutoEval-G's graph dataset discrepancy representation $f'_{\text{linear}}$, which measures the graph distribution gap between the observed training graph $\mathcal{S}$ and each meta-graph $g_{\text{meta}} \in \mathcal{G}_{\text{meta}}$ with their node embeddings from the well-trained $\text{GNN}_{\mathcal{S}}$ as:

$$f'_{\text{linear}} = \text{MMD}(\mathcal{S}, g_{\text{meta}}) = \text{MMD}(\mathbf{Z}_{\mathcal{S}}^*, \mathbf{Z}_{\text{meta}}^{(i,*)}). \tag{14}$$

where $\mathbf{Z}_{\mathcal{S}}^* = \text{GNN}_{\mathcal{S}}^*(\mathbf{X}, \mathbf{A})$, and $\mathbf{Z}_{\text{meta}}^{(i,*)} = \text{GNN}_{\mathcal{S}}^*(\mathbf{X}_{\text{meta}}^i, \mathbf{A}_{\text{meta}}^i)$.

# D Well-trained GNN Model Details

In this section, we provide the details of well-trained GNN models in terms of their architecture parameters and training details.

For all GNN and MLP models, the default settings are : (a) the number of layers is 2; (b) the hidden feature dimension is 128; (c) the output feature dimension before the softmax operation is 16. The hyperparameters of training these GNNs and MLP are shown in Table A2. The five seeds for training each type of models (GCN, GraphSAGE, GAT, GIN, MLP) are $\{0, 1, 2, 3, 4\}$, and the node classification performance (accuracy) on the test set of each observed training graph and the ground-truth accuracy on the unseen test graphs without labels are shown in Table A3, Table A4, Table A5, Table A6, and Table A7, respectively.

Table A3: Node classification results in accuracy(%) of GCN with different seeds in terms of the test set of the observed graph and unseen test graphs.

| GCN | Observed ACMv9 | | | Observed Citationv2 | | | Observed DBLPv8 | | |
|---|---|---|---|---|---|---|---|---|---|
| Seeds | $A_{test}$ | GT A→C | GT A→D | $C_{test}$ | GT C→A | GT C→D | $D_{test}$ | GT D→A | GT D→C |
| 0 | 83.13 | 45.51 | 63.88 | 88.98 | 46.86 | 60.18 | 97.58 | 70.81 | 57.65 |
| 1 | 83.74 | 49.01 | 68.09 | 88.45 | 40.76 | 61.60 | 98.30 | 68.43 | 57.90 |
| 2 | 82.73 | 55.08 | 69.29 | 88.45 | 45.51 | 60.95 | 98.57 | 70.04 | 53.36 |
| 3 | 83.40 | 48.89 | 69.85 | 89.09 | 46.09 | 60.58 | 98.57 | 68.11 | 52.85 |
| 4 | 83.87 | 55.63 | 68.75 | 88.24 | 37.25 | 58.62 | 98.66 | 70.61 | 58.37 |

Table A4: Node classification results in accuracy (%) of GraphSAGE with different seeds in terms of the test set of the observed graph and unseen test graphs.

| SAGE | Observed ACMv9 | | | Observed Citationv2 | | | Observed DBLPv8 | | |
|---|---|---|---|---|---|---|---|---|---|
| Seeds | $A_{test}$ | GT A→C | GT A→D | $C_{test}$ | GT C→A | GT C→D | $D_{test}$ | GT D→A | GT D→C |
| 0 | 82.32 | 45.49 | 63.54 | 88.98 | 50.59 | 59.90 | 98.66 | 42.24 | 42.27 |
| 1 | 83.54 | 48.97 | 69.67 | 88.14 | 48.87 | 58.03 | 98.84 | 39.41 | 35.21 |
| 2 | 83.67 | 52.56 | 71.15 | 88.45 | 41.17 | 54.61 | 98.39 | 45.82 | 55.99 |
| 3 | 82.79 | 48.19 | 66.22 | 88.03 | 42.20 | 53.96 | 98.48 | 31.30 | 27.25 |
| 4 | 82.05 | 47.04 | 66.67 | 88.35 | 37.46 | 55.70 | 98.84 | 36.99 | 35.14 |

Table A5: Node classification results in accuracy (%) of GAT with different seeds in terms of the test set of the observed graph and unseen test graphs.

| GAT | Observed ACMv9 | | | Observed Citationv2 | | | Observed DBLPv8 | | |
|---|---|---|---|---|---|---|---|---|---|
| Seeds | $A_{test}$ | GT A→C | GT A→D | $C_{test}$ | GT C→A | GT C→D | $D_{test}$ | GT D→A | GT D→C |
| 0 | 83.00 | 44.84 | 72.53 | 87.82 | 41.55 | 57.94 | 97.85 | 64.10 | 51.77 |
| 1 | 82.73 | 44.18 | 57.08 | 88.67 | 48.80 | 60.40 | 98.84 | 57.03 | 42.18 |
| 2 | 81.65 | 45.07 | 69.59 | 88.77 | 45.74 | 59.04 | 98.48 | 39.65 | 63.99 |
| 3 | 82.79 | 43.82 | 60.18 | 88.98 | 41.48 | 61.03 | 98.93 | 62.33 | 58.43 |
| 4 | 81.71 | 42.04 | 57.62 | 87.50 | 37.04 | 60.06 | 98.30 | 63.64 | 52.64 |

Table A6: Node classification results in accuracy (%) of GIN with different seeds in terms of the test set of the observed graph and unseen test graphs.

| GIN | Observed ACMv9 | | | Observed Citationv2 | | | Observed DBLPv8 | | |
|---|---|---|---|---|---|---|---|---|---|
| Seeds | $A_{test}$ | GT A→C | GT A→D | $C_{test}$ | GT C→A | GT C→D | $D_{test}$ | GT D→A | GT D→C |
| 0 | 82.19 | 50.37 | 74.94 | 88.24 | 33.35 | 59.47 | 98.48 | 35.13 | 40.78 |
| 1 | 82.52 | 43.12 | 51.58 | 86.97 | 35.40 | 64.83 | 66.70 | 35.37 | 28.14 |
| 2 | 82.86 | 42.93 | 58.21 | 81.57 | 39.28 | 65.04 | 98.93 | 53.91 | 30.56 |
| 3 | 82.46 | 47.93 | 69.79 | 87.92 | 44.59 | 69.72 | 63.47 | 30.61 | 21.53 |
| 4 | 82.39 | 47.04 | 78.13 | 88.56 | 40.23 | 65.72 | 98.93 | 59.39 | 45.56 |

Table A7: Node classification results in accuracy (%) of MLP with different seeds in terms of the test set of the observed graph and unseen test graphs.

| MLP | Observed ACMv9 | | | Observed Citationv2 | | | Observed DBLPv8 | | |
|---|---|---|---|---|---|---|---|---|---|
| Seeds | $A_{test}$ | GT A→C | GT A→D | $C_{test}$ | GT C→A | GT C→D | $D_{test}$ | GT D→A | GT D→C |
| 0 | 71.59 | 33.28 | 38.17 | 74.89 | 34.48 | 37.25 | 62.67 | 59.18 | 51.41 |
| 1 | 70.18 | 33.47 | 39.05 | 74.58 | 35.30 | 38.62 | 63.12 | 54.44 | 43.92 |
| 2 | 70.11 | 42.65 | 43.67 | 74.36 | 34.74 | 39.12 | 63.12 | 53.74 | 42.59 |
| 3 | 68.62 | 35.12 | 39.71 | 72.56 | 33.91 | 37.41 | 62.94 | 54.28 | 44.31 |
| 4 | 69.23 | 35.95 | 42.33 | 74.58 | 34.41 | 39.82 | 64.37 | 54.08 | 43.88 |

# E    More Experimental Analysis and Results.

## E.1    DiscGraph Statistics

As a vital component of our proposed two-stage GNN model evaluation framework, DiscGraph set captures wide-range and diverse graph data distribution discrepancies. By fully exploiting the latent node embeddings and node class predictions from well-trained GNNs, the DiscGraph set involves representative node attributes incorporating diverse discrepancies through a discrepancy measurement function, leading to effective instructions for the training of `GNNEvaluator`. Hence, in the following, we show the statistical information of our derived DiscGraph set in Table A8. Note that, each generated DiscGraph set is related to a specific dataset and a specific well-trained GNN, as it considers the outputs of node embeddings and predictions on a well-trained GNN trained by a specific graph dataset.

Table A8: Statistics of the proposed DiscGraph in terms of GNNs on ACMv9, Citationv2, DBLPv8, where the accuracy label distributions calculate average minimum and maximum node classification accuracy on GNNs with five random seeds with standard deviations (std).

| Statistics | | DiscGraph-ACMv9 | DiscGraph-Citationv2 | DiscGraph-DBLPv8 |
|---|---|---|---|---|
| #Num of Graphs ($K$) | | 400 | 400 | 400 |
| #Avg. Num of Nodes | | 2105 | 1340 | 1585 |
| #Avg. Num of Edges | | 1595 | 927 | 1205 |
| | GCN | $39.95_{\pm0.00}$ \| $80.45_{\pm1.40}$ | $13.89_{\pm7.82}$ \| $85.36_{\pm1.00}$ | $17.53_{\pm0.57}$ \| $91.95_{\pm0.56}$ |
| | SAGE | $39.95_{\pm0.00}$ \| $77.76_{\pm2.22}$ | $21.29_{\pm7.28}$ \| $82.43_{\pm0.54}$ | $19.07_{\pm3.52}$ \| $77.27_{\pm2.03}$ |
| Accuracy Label Distributions | GAT | $39.50_{\pm1.01}$ \| $79.67_{\pm0.63}$ | $19.53_{\pm12.27}$ \| $86.13_{\pm0.47}$ | $19.58_{\pm1.34}$ \| $87.26_{\pm3.99}$ |
| (Min. \| Max.$_{\pm\text{std}}$) | GIN | $28.50_{\pm10.15}$ \| $76.06_{\pm2.27}$ | $13.48_{\pm8.24}$ \| $81.50_{\pm2.91}$ | $19.06_{\pm5.18}$ \| $63.63_{\pm16.83}$ |
| | MLP | $39.92_{\pm0.04}$ \| $71.91_{\pm1.19}$ | $33.65_{\pm7.08}$ \| $77.41_{\pm0.75}$ | $17.71_{\pm0.39}$ \| $64.68_{\pm0.41}$ |

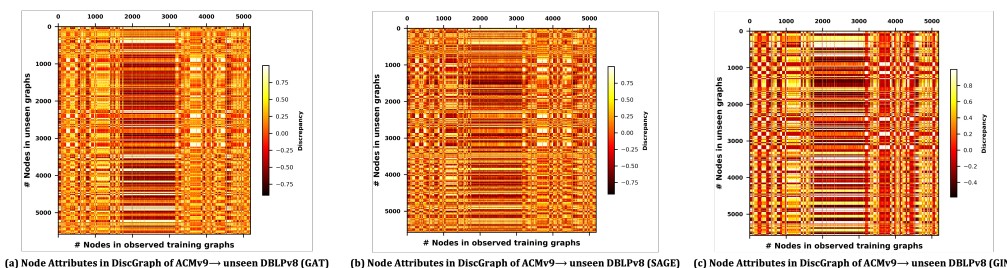

(a) Node Attributes in DiscGraph of ACMv9→ unseen DBLPv8 (GAT)    (b) Node Attributes in DiscGraph of ACMv9→ unseen DBLPv8 (SAGE)    (c) Node Attributes in DiscGraph of ACMv9→ unseen DBLPv8 (GIN)

Figure A1: Visualizations on discrepancy node attributes in the proposed DiscGraph set for (a) GAT, (b) GraphSAGE, and (c) GIN model evaluation. Darker color denotes a larger discrepancy.

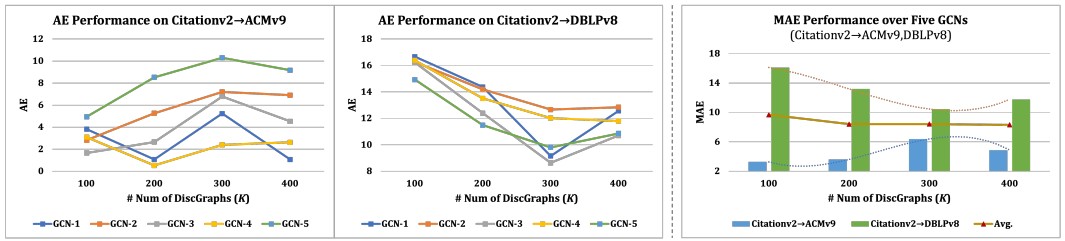

Figure A2: Effects of different numbers of DiscGraphs ($K$) for `GNNEvaluator` training with Absolute Error (AE) for each GCN model under C→A and C→D cases.

### E.2 More Visualization Results of Discrepancy Node Attributes

In Fig. A1, we present more visualization results on discrepancy node attributes in the proposed DiscGraph set for different GNN models, *i.e.*, (a) GAT, (b) GraphSAGE, and (c) GIN, under ACMv9→DBLPv8 case.

As can be observed, for different GNNs, the node attributes in the proposed DiscGraph set show significant differences, denoting the effectiveness of the proposed discrepancy measurement function, which could capture model-specific discrepancy representations effectively. And such discrepancy representations could instruct our proposed `GNNEvaluator` to learn graph data distribution discrepancies with different well-trained GNNs for accuracy regression.

### E.3 More Results on The Number of DiscGraphs

We provide more results for illustrating the effects of different numbers of discGraphs for training the proposed `GNNEvaluator` in Fig. A2. Along with the MAE results reported in Fig. 5 in the main manuscript, we report the Absolute Error (AE) results for each GCN model under C→A and C→D cases. Lower AE denotes better performance. It can be observed that different GCN models have different appropriate $K$ values for `GNNEvaluator` training, but they show similar trends for each unseen test graph in the left and the middle figures. For instance, the proposed `GNNEvaluator` has highest AE when it is trained with $K = 100$ DiscGraphs, and comparable AEs with $K = 300$ and $K = 400$ for evaluating all five GCNs under the C→D case. Nevertheless, in general, as shown in the red 'Avg.' line of the right figure, the performance of the proposed `GNNEvaluator` would not significantly vary along the number of DiscGraphs when evaluating all GCNs that are well-trained on Citationv2 dataset and served on unseen DBLPv8 dataset.

