# OpenReview forum: "GNNEvaluator: Evaluating GNN Performance On Unseen Graphs Without Labels"
_NeurIPS.cc/2023/Conference — NeurIPS 2023 poster_

### Official Review · Reviewer_ZAqu · 2023-06-08

**Soundness:** 3 good
**Presentation:** 3 good
**Contribution:** 4 excellent
**Rating:** 6
**Confidence:** 4

**Summary:**

This paper addresses the problem of evaluating the performance of well-trained Graph Neural Networks (GNNs) on unseen test graphs without ground-truth labels. Traditional evaluation methods that rely on annotated datasets are not applicable in real-world scenarios where test graphs are unlabeled. The paper proposes a two-stage evaluation framework: DiscGraph set construction and GNNEvaluator training and inference. The framework effectively captures the distribution discrepancies between training and test graphs, models graph structural discrepancies, and estimates node classification accuracy without labeled data. Experimental results indicate the success of the proposed method.

**Strengths:**

1. I really appreciate the practical value of this paper since it often requires a long time for evaluation a model in the real-world scenario. The paper introduces the novel problem of evaluating GNN models on unseen test graphs without labels, which is essential for real-world GNN deployment and serving.

2. The proposed framework first constructs a set of meta-graphs to simulate potential unseen test graphs, captures graph data distribution discrepancies, and models graph structural discrepancies using latent node embeddings and node class predictions. The GNNEvaluator is then trained to estimate node classification accuracy based on the representative DiscGraph set.

3. The paper evaluates the proposed method on real-world unlabeled test graphs and demonstrates its effectiveness in estimating node classification accuracy.

**Weaknesses:**

1.  In section 3.3, the author introduces extracting a seed sub-graph Sseed from the observed training graph S, Nonetheless, the details about how to extract the seed subgraph lack of introduction.

2.  The estimation of the represented discrepancy is too simple. I am concerned that not be generalized to larger graph in more domains, especially those larger datasets, e.g., ogbn-arxiv, ogbn-products. Those graphs show more diverse patterns and the proposed method may not work in this scenario.

3. The graph argumentation method is to provide sufficient quantity. Nonetheless, the graph argumentation algorithms seem to only disturb the graph very small. I am wondering how those methods can be generalized to other scenarios with larger gap. Moreover, I am wondering whether it can help to predict the performance on the model trained with those graph argumentation techniques, e.g., dropedge.

4.  I am wondering why the GNN evaluator can have the transferability, e.g., the learned GNN evaluator on SAGE can also help to predict the performance of the GCN.

**Questions:**

See the above weakness

**Limitations:**

The authors have adequately addressed the limitations

---

> ### Author Rebuttal · Authors · 2023-08-09
>
> **Response to Reviewer ZAqu**
>
> We sincerely appreciate your valuable suggestions and comments on our work, and we are pleased to learn that the practical value of our proposed GNNEvaluator is positively identified by the reviewer. The following are our detailed responses to the reviewer’s thoughtful comments. We are expecting these could be helpful in answering your questions.
>
> **W1: How to extract the seed subgraph**:
> The seed subgraph is sampled from the observed graph S. For instance, in our experiments, for the observed ACM dataset, we use its 30% nodes to construct the subgraph as the seed subgraph. Thanks for your suggestion and we will add these details to the final version.
>
> **W2: Generalize to the larger graph in more domains for represented discrepancy estimation**:
> For GNN model evaluation, our core idea is to construct a set of discrepancy meta-graphs for modeling complex and diverse graph discrepancies. And in each discrepancy meta-graph, we measure the node-level represented discrepancy by fully leveraging the output node embeddings from well-trained GNN models.
> Hence, for larger graphs in more diverse domains (e.g., ogbn-arxiv, ogbn-products), a possible solution is to create more discrepancy meta-graphs to comprehensively cover more diverse patterns with various represented discrepancies.  However, it's worth noting that constructing a thorough, extensive, and diverse discrepancy meta-graph set for more complex and larger graph domains is still a challenging task, and we are happy to further explore this point in the future.
>
> **W3-1: The graph argumentation algorithms seem to only disturb the graph very small, and how those methods can be generalized to other scenarios with larger graph gaps**:
> We introduce 4 types of graph augmentation methods, and each has 100 random disturbations when the disturbing rate is sampled from a uniform distribution (0,1).  For example, the total 400 disturbations could make the GCN’s node classification accuracy varies in a minimum 17.53% to a maximum 91.95% on DBLPv8 dataset. These results illustrate that the graph argumentation methods could disturb the graph in a relatively large range. Hence, it could generalize to other scenarios with larger graph gaps by involving more graph augmentation methods and more diverse disturbing rates.
>
> **W3-2: Whether GNNEvaluator can help to predict the performance on the model trained with those graph argumentation techniques, e.g., dropedge**:
> Yes, our proposed GNNEvaluator can be used to predict the performance of the model trained with graph argumentation techniques. That is because, in the model evaluation stage, the well-trained model is fixed and our proposed GNNEvaluator only leverages its output node embeddings and node class predictions. That means, even the model trained with graph argumentation, our proposed GNNEvaluator could still directly leverage its outputs to predict its performance for its evaluation.
>
> **W4: Why the GNN evaluator can have the transferability**:
> We would like to clarify the potential misunderstanding here. In our experiments, the “transferability” is reflected in the well-trained GNNs* (e.g, GAT model trained on ACMv9 but used for inference on DBLPv8) *across datasets*. And we use our proposed GNNEvaluator to estimate these well-trained GNNs* transferred node classification accuracy. Hence, our proposed GNNEvaluator should NOT be expected to have the transferability *across models*, since the GNNEvaluator is a model evaluator, so it should be model-specific and driven by different GNN models. If it is learned on SAGE, our whole evaluation process is based on the SAGE’s outputs of node embeddings and class predictions, so that it can only be used for evaluating SAGE, rather than GCN.

---

> > ### Comment · Reviewer_ZAqu · 2023-08-10
> > **Response**
> >
> > Thanks for your response. I would like to keep the score since this method is still not of practical use without results on larger datasets.

---

> > > ### Author Response · Authors · 2023-08-11
> > > **Reply to Reviewer ZAqu's response to our rebuttal**
> > >
> > > Dear Reviewer ZAqu,
> > >
> > > Thank you for your response to our rebuttal. We are glad our efforts have addressed most of your questions and concerns, and we really appreciate your support for this new research domain of “GNN model evaluation”.
> > >
> > > In this very early phase of exploration, we also hold the practical application of GNN model evaluation in high regard. Our dedicated efforts are centered around *shaping this research problem to fit real-world, practical scenarios involving unseen and unlabeled test graphs*. And our GNNEvaluator stands as **the first feasible solution**, serving to enlighten GNN model developers and users about the potential performance of well-trained models.
> > >
> > > We firmly believe that within this emerging realm, numerous captivating questions and opportunities await thorough exploration. We're dedicated to further advancing this field, inspiring future research that delves into comprehensive GNN model evaluations across extensive graph data scales and properties!

---

### Official Review · Reviewer_jBkz · 2023-06-30

**Soundness:** 2 fair
**Presentation:** 3 good
**Contribution:** 3 good
**Rating:** 6
**Confidence:** 4

**Summary:**

The paper presents a novel problem called GNN model evaluation, aiming to assess the performance of a Graph Neural Network (GNN) on unseen graphs without labels. The authors propose a two-stage GNN model evaluation framework, which includes DiscGraph set construction and GNNEvaluator training and inference. The DiscGraph set is designed to capture diverse graph data distribution discrepancies using a discrepancy measurement function that exploits GNN outputs. The GNNEvaluator, composed of a GCN architecture and an accuracy regression layer, learns to estimate node classification accuracy with effective supervision from the DiscGraph set. The method demonstrates effectiveness in evaluating GNN performance on real-world, unlabeled test graphs, achieving low errors compared to ground-truth accuracy.

**Strengths:**

This paper is the first to evaluate GNN performance on out-of-distribution (OOD) graphs, which may inspire future studies. The meta-graph set construction and the three characteristics are intriguing and useful. Additionally, the proposed method demonstrates significant performance improvements over the baselines.

**Weaknesses:**

This paper has a few areas that could be improved upon. I would consider raising the score if the authors addressed the first two points.
1. The paper lacks a discussion of the use cases for the proposed method, considering the prediction error remains relatively high (Please refer to Question 1).
2. The experiments involve only four datasets, and there is a lack of comprehensive study on node classification in different settings (Please refer to Question 2).
3. The paper does not provide ablation studies for the predictor in GNNEvaluator (Please refer to Question 3).

**Questions:**

To enhance the manuscript, the authors may consider addressing the following points:

1. Although the proposed method outperforms the baselines, the prediction error remains high. For example, in ACMv9 -> DBLPv8, the performance of GCN, SAGE, GAT, GIN, and MLP are 45.51%, 45.49%, 44.84%, 50.37%, and 33.28%, respectively, and the evaluation errors are 4.85%, 4.11%, 12.23%, 10.14%, 22.20%, respectively. The relative evaluation errors for various GNNs are quite significant. With such high relative errors, it is challenging for GNNEvaluator to distinguish the performance of any two GNNs. In light of this, what are the practical applications of the proposed method? Including a discussion of compelling real-world applications could make this paper more appealing.

2. While the paper focuses on node classification settings, which is an important application of GNNs, the experiments only evaluate inductive node classification on citation networks. To make the experiments more comprehensive, I suggest that the authors investigate (1) inductive node classification on heterophilous graphs, and (2) transductive node classification.

3. The paper employs a two-layer GCN as a predictor, but detailed ablation studies are missing. For example, (1) what is the variance of the GCN predictor? (2) could a simpler model (e.g., SGC) or a more complex model achieve better performance? It would be beneficial to include such analysis in the paper.

**Limitations:**

Yes

---

> ### Author Rebuttal · Authors · 2023-08-09
>
> **Response to Reviewer jBkz**
>
> Thanks for your insightful and constructive review of our work. We especially appreciate your interest in more exploration experiments of our proposed GNNEvaluator, and we are encouraged to know that our efforts on "the meta-graph set construction and the three characteristics" have been recognized. Following are our responses, and we are expecting these could help answer your questions.
>
> **Q1-What are the practical applications of the proposed method?**:
> *Recap*: Our proposed GNNEvaluator aims to provide a feasible solution to know the potential performance of existing well-trained GNNs on real-world unseen graphs “without labels”, by directly predicting accuracy. *Application*: The most straightforward practical application is our GNNEvaluator aids in selecting relatively well-performing GNNs from model collections, enabling us to have confidence in their performance on new, unseen, and unlabeled graphs.  While relatively high prediction errors in mentioned cases (ACMv9 $\rightarrow$Citationv2, there are 6 classes) in Table 1 of our main submission, we list the rank of different GNN models between the Ground-Truth ACC (%) and our GNNEvaluator predicted ACC (%) in following Table.Re-jBkz-1.  We could observe that our predicted ACC could have a consistent ranking of different GNN models even with different prediction errors.  In this case, we can still place greater trust in the "GIN" model although there is still a gap between our prediction and the Ground-Truth. Due to the unexplored research field on "GNN model evaluation", we would like to emphasize that our proposed GNNEvaluator achieves this goal by predicting performance on unseen and unlabeled data with classification accuracy. While this is still an early exploration,  our experimental results could shed light on the promising potential for practical model selection applications. We are committed to continuous efforts, and hopefully inspiring future research to reduce the prediction error and further explore this research direction.
>
> **Table.Re-jBkz-1. Rank comparison of different  GNN models between the Ground-Truth ACC (%) and our GNNEvaluator predicted ACC (%).**
> | Models | Prediction error | GT target ACC (%) | Ours predict ACC (%) | Rank-GT-ACC | Rank-Ours-ACC |
> | :---:| :---: | :---: | :---: | :---: | :---: |
> | GCN | 10.09 | 45.51 | 55.61 | 2 | 2 |
> | SAGE | 7.19 | 45.49 | 55.44 | 3 | 3 |
> | GAT | 9.11 | 44.84 | 53.94 | 4 | 4 |
> | GIN | 6.11 | 50.37 | 56.49 | 1 | 1 |
>
> **Q2: Investigate (1) experiments on heterophilous graphs, and (2) transductive node classification.**
> For **(1) experiments on heterophilous graphs**, we have conducted experiments on two typical webpage heterophilous graphs, Cornell and Texas with heterophily degrees of 0.11 and 0.16, respectively. The MAE experimental results are shown in **Table.Re 1 of the response PDF file**, with Cornell$\rightarrow$Texas case and Texas$\rightarrow$Cornell case (the lower, the better). All detailed settings are consistent with our main submission. As can be observed, our proposed GNNEvaluator generally achieves superior performance for all GNN models than other baselines, further demonstrating the effectiveness of our method.
>
> For **(2) transductive node classification**, it means the test nodes can be seen in the model training process by leveraging their neighbor structure information even without using node labels.
> However, this setting is not aligned with the proposed GNN model evaluation scenario, where the unlabeled test graphs are usually unseen for real-world practical applications. For instance, consider a citation network, where new paper nodes continually emerge over time. These new nodes are strictly unavailable during the model's training phase, making the transductive setup unfeasible.
>
> **Q3-1: What is the variance of the GCN predictor?**:
> In the following Table.Re-jBkz-2, we list the results of Mean Absolute Error (MAE)$\pm$standard deviation (STD) of different GNN models across five runs to illustrate the variance of our proposed GNNEvaluator (GCN predictor). Compared with baseline method AutoEval-G, our proposed method could achieve better MAE results with comparable even better variance with smaller STD on some cases, for example, for Citationv2$\rightarrow$DBLPv8 case on GAT model, we have only 0.76 variance, better than 3.17 variance of  AutoEval-G.
>
> **Table.Re-jBkz-2. Mean Absolute Error (MAE)$\pm$standard deviation (STD) of different GNN models for Citationv2$\rightarrow$DBLPv8.**
> | Citationv2$\rightarrow$DBLPv8 | GCN | SAGE | GAT | GIN | MLP |
> | :--- | :---: | :---: | :---: | :---: | :---: |
> | AutoEval-G | 2.57$\pm$2.24 | 16.52$\pm$1.88 | 6.96$\pm$3.17 | 19.20$\pm$15.36 | 32.24$\pm$1.92 |
> | GNNEvaluator (**Ours**) | 11.64$\pm$5.35 | 7.02$\pm$1.38 | 5.58$\pm$0.76 | 6.46$\pm$5.17 | 22.87$\pm$2.42 |
>
>
> **Q3-2: Could a simpler model (e.g., SGC) or a more complex model achieve better performance?**:
> We tested the effectiveness of different backbone architectures in terms of GNN predictor in **Table.Re 2 of the response PDF file**. Compared with the GCN-backboned evaluator used in our main submission, we evaluate the simper model SGC-backboned evaluator and a more complex GPRGNN-backboned evaluator with a complex aggregation scheme.
> According to the results, the simpler SGC-backboned evaluator fails to achieve satisfactory performance when it is too simple to effectively model complex and diverse graph discrepancies. Although the complex GPRGNN-backboned evaluator could have a relatively good performance in certain SAGE evaluation (although still comparable to our GCN-backboned evaluator), it could not achieve consistently better performance for all GNNs.  In contrast, our proposed GCN-backboned evaluator could be a good and general choice with relatively consistent performance, considering the diversity of real-world application scenarios. We will add these discussions to the final version.

---

### Official Review · Reviewer_3iA3 · 2023-07-07

**Soundness:** 4 excellent
**Presentation:** 4 excellent
**Contribution:** 4 excellent
**Rating:** 7
**Confidence:** 3

**Summary:**

This paper studies the problem of evaluating a GNN model, to estimate the accuracy on unseen unlabeled data for a well trained model. The goal of this paper is not to improve the generalization error of a GNN model, rather estimate the well-trained model error. This is a novel track for model generalization error, where before only studied in Euclidian data. It contains two parts, DiscGraph, that extracts a set of meta-graphs from the original training graph to simulate and capture the discrepancies of diverse graph data distributions for the aforementioned unseen test graphs. In addition, it proposes GNNEvaluator, a model to estimate the trained model accuracy  on the unseen test data set.

**Strengths:**

This is a very well written paper, and it is aiming at a new domain of research for GNNs. The proposed idea is very well presented, in addition to detailed analysis of each part of the GNNEvaluator. Due to lack of related work for GNN model, the authors introduced and adapt recent works from recent CNN algorithm. Overall, I found this paper very novel.

**Weaknesses:**

I found the paper very interesting, however, the experimental studies is very limited (understandable due to the novelty of the problem). However it is not very convincing to see the result only on one set of application (for limited number of labeled class).
Also it would be beneficial to see the performance on different architectures (deeper GNN) as right now the GNNEvaluator only uses 2 layers GCN.

**Questions:**

- Related to the problem mentioned above, how we can be certain about the number of DiscGraph generated for first stage?
- What is the time and memory requirement for these additional stages for GNN training? Given the limited and small evaluated graphs, how scalable is current approach?

---

> ### Author Rebuttal · Authors · 2023-08-09
>
> **Response to Reviewer 3iA3**
>
> We sincerely appreciate your thoughtful review of our paper. We are so encouraged by your recognition of the “new domain of research for GNNs” on GNN model evaluation, and this means a lot to advancing the GNN model inference and deployment in real-world applications. We have carefully considered your comments and suggestions, and the following are our detailed responses. We are expecting these could be helpful in answering your questions.
>
> **W1:  The experimental studies is very limited (understandable due to the novelty of the problem), and clarification of the results on  limited number of labeled class application**:
> Thanks for your valuable comments. Due to the complexity and diversification of unobserved and unlabeled test graphs in real-world GNN model evaluation, in our experiments, we have tried our best to involve more experimental cases, for instance, testing 5 typical GNN models on 6 transfer inference cases over 3 datasets,  with totally 5$\times$6=30 evaluation scenarios (Table 1-3 in the main submission); as well as ablation studies and more discussion (in Fig. 3-5 of the main submission and Table A1-A8, Fig. A1-A2 in Appendix). We are committed to continuous efforts for more comprehensive future studies to advance and explore this new research topic of GNN model evaluation.
>
> For the label class setting, we assume the observed training graph and the potential unseen and unlabeled test graph has the same number of label classes, for instance, both C-classes. This setting could align with the ability of current well-trained GNN models to be evaluated. When the to-be-evaluated GNN model is well-trained with C-classes on the observed training graph, it can not be directly used for inferring the unlabeled and unseen test graphs with unknown label classes (for example, C+1 classes) since its model architecture has only C-class prediction outputs.
> We appreciate your insightful comments, and we believe this is a very interesting question.
> As mentioned at the end of our main submission (Lines 359-361), we are happy to further explore this in the future.
>
> **W2: It would be beneficial to see the performance on different architectures (deeper GNN) as right now the GNNEvaluator only uses 2 layers of GCN**:
> As shown in following Table.Re-3iA3-1, we compared the results of GCN-Predictor (3-layers) with GCN-Predictor (**2-layers, ours**) on DBLPv8$\rightarrow$ACMv9 (D$\rightarrow$A) and DBLPv8$\rightarrow$Citationv2(D$\rightarrow$C) on GCN, SAGE, and GAT evaluations. As can be observed, the 2-layer setting in our main submission still achieves generally better performance compared with 3-layer GCN evaluator.
> The reason behind this might be that the deeper GNN might have an over-smoothing issue when the node features might be too similar to discriminate. Thanks for your thoughtful suggestions and we will add this to the final version.
>
> **Table.Re-3iA3-1. Mean Absolute Error (MAE) performance on different GNNEvaluator layers (the lower, the better).**
> | Datasets |  | D$\rightarrow$A |  |  |
> | :--- | :--- | :--- | :--- | :--- |
> | Models | GCN | SAGE | GAT | *Avg.* |
> | GCN-Predictor (3-layers) | 4.85 | 11.89 | **5.65** |7.46 |
> | GCN-Predictor (**2-layers, ours**) | **2.46** | **10.27** | 6.94 | **6.56** |
> | **Datasets** |  | **D$\rightarrow$C** |  |  |
> | Models | GCN | SAGE | GAT | *Avg.* |
> | GCN-Predictor (3-layers) | 17.57 | 11.97 | 5.44 |11.66 |
> | GCN-Predictor (**2-layers, ours**) | **11.68** | **7.83** | **3.97** | **7.83** |
>
> **Q1: How we can be certain about the number of DiscGraph generated for first stage?**:
> At the first stage, we can not be very certain about the number of DiscGraphs, we can only study the effects by empirical experiments as mentioned in Sec. 4.5 of our main submission. As can be observed, the results slightly change with the number of DiscGraph, but not too sensitive over certain numbers (over 200) on average.
>
> **Q2-1: What is the time and memory requirement for these additional stages for GNN training?**:
> We test the run time and memory usage on a single NVIDIA GeForce RTX 3080 GPU on GCN model evaluation that is well trained on DBLPv8 dataset, and the running time is only 44.60 seconds per 10 training epochs and the overall dynamic memory usage is only 1990MB. Thanks for your valuable questions, and we will add this to our final version.
>
> **Q2-2 Given the limited and small evaluated graphs, how scalable is current approach?**:
> The largest graph (ACMv9) used in our experiments contains 7k nodes with 11k edges, and it can still be modeled with the 2-layer GCN evaluator used in our main submission.
> For the larger scale of unseen test graphs in the real world, we might consider using graph sampling techniques or changing the backbone of the GNN evaluator to make it adapt to super-large graphs, such as GraphSage backboned evaluator, or GraphSAINT backboned evaluator. Thanks for your insightful question and we are happy to explore more on this point in the future.

---

### Official Review · Reviewer_nFGD · 2023-07-07

**Soundness:** 3 good
**Presentation:** 3 good
**Contribution:** 3 good
**Rating:** 7
**Confidence:** 4

**Summary:**

This work proposes a new research problem named GNN model evaluation, to evaluate well-trained GNNs on observed training graphs for testing on real-world unobserved test graphs without labels. To achieve this goal, this work (1)constructs a DiscGraph set to model the distribution differences of graph datasets, and (1) designs a GNNEvaluator to learn on the constructed DiscGraph set and directly output the overall node classification accuracy by a regression model.
Experimental results on some shifted-distribution graph datasets verify the proposed method and questions' effectiveness.


**Strengths:**

(1) This work first defines the GNN model evaluation problem, which is a very practical problem in real-world gnn model deployment and selection for user sides. This work has good originality and explores a new research problem for real-world GNN deployment and applications. It could inspire some interesting explorations in terms of GNN model service and deployment.


(2) The challenges of the GNN model evaluation, the corresponding technical solutions proposed by this work and experimental settings are clearly clarified. And the DiscGraph set constructed for capturing the graph dataset discrepancy is novel with sufficient quantity, represented discrepancy, and known accuracy. The calculated distance of two graph dataset distribution as the node attributes sound rational.
And a new trained GNN regressor on this constructed DiscGraph set for predicting GNN's classification result over the whole dataset is reasonable.

(3) The experimental setting could align with the real-world application scenario, and the result parts could support this work's claims. Appropriate ablation study, DiscGraph set analysis, and hyper-parameter analysis provides adequate information for verifying the method's effectiveness with small errors from the ground truth accuracy.

(4) This work is well-structured and clearly written with logic.



**Weaknesses:**

See Questions

**Questions:**

(1) How to obtain the seed graph for augmentation to generate the meta-graph? it would be better to show the difference between the observed graph used for training and the created meta-graph for constructing the DiscGraph set.

(2) Do the results in each table (tables 1 2 and 3), share the same trained models? for instance, in table 2, a gcn model is well-trained on citationv2 but evaluated the same model on both acmv9 and dblpv8?

(3) The authors should provide more experimental analysis and discussions to demonstrate why such a method design could benefit gnn model evaluation compared with other baselines. Besides, there might be a typo in table 1 first six columns? should be acmv9 to citationv2 dataset?

---

> ### Author Rebuttal · Authors · 2023-08-09
>
> **Response to Reviewer nFGD**
>
> We sincerely appreciate your thoughtful review of our paper. We are glad to hear that you recognize the significance of GNN model evaluation problem proposed by our work. We have carefully considered your comments and suggestions, and the following are our detailed responses. We are expecting these could be helpful in answering your questions.
>
> **Q1-1: How to extract the seed subgraph**:
> The seed subgraph is sampled from the observed graph S. For instance, in our experiments, for the observed ACM dataset, we use its 30% nodes to construct the subgraph as the seed subgraph. Thanks for your suggestion and we will add these details to the final version.
>
> **Q1-2:  Show the difference between the observed graph and the constructed meta-graph**:
> As mentioned in Sec.E.2 and Fig. A1 of our Appendix, we have provided the visualizations on discrepancy node attributes in the proposed DiscGraph set for GAT, GraphSAGE, GIN model evaluations. As can be observed, for different GNNs, the node attributes in the proposed DiscGraph set show significant differences, denoting that our proposed discrepancy measurement function could effectively capture model-specific discrepancy representations effectively.
>
> **Q2:  Do the results in each table (tables 1 2 and 3), share the same trained models**:
> Yes, the results in each table (tables 1 2 and 3) share the same trained models. For the GNN model evaluation problem, the well-trained GNN models on observed training graphs should be fixed on the stage of unlabeled real-world graph inference. Hence, a GCN model well-trained on Citationv2 is evaluated the same model on both ACMv9 and DBLPv8.
>
> **Q3: More experimental analysis and discussions to demonstrate why such a method design could benefit GNN model evaluation compared with other baselines**:
> The good performance of our proposed GNNEvaluator that benefits GNN model evaluation could attribute to the following reasons according to our experimental results: (1) compared with other methods, we comprehensively simulate and capture the discrepancies of diverse graph data distributions within the constructed DiscGraph set (Results in Fig.3, 4, 5 in the main submission); (2) we design a GNNEvaluator to directly and effectively estimate node classification accuracy of unseen and unlabeled real-world graphs (Results in Table 1, 2, 3  in the main submission). Thanks for your valuable suggestions and we will add more discussions and analysis in the final version. In the first six columns of Table.1, it should be acmv9 to citationv2 dataset, and we will correct this typo in the final version.

---

### Author Rebuttal · Authors · 2023-08-09

**Common response to all reviewers**:

We thank all reviewers for their thorough review and valuable suggestions. We are delighted that our contributions have been positively acknowledged, including:

**(1) Novel research question for new domain exploration of GNN model evaluation problem ( @All Reviewers!)**;

**(2) Practical application value for real-world GNN deployment and serving (Reviewer nFGD, Reviewer ZAqu)**;

**(3) Novel, intriguing, and useful discrepancy meta-graph set construction (Reviewer nFGD, Reviewer jBkz, Reviewer ZAqu)**;

**(4) Reasonable GNNevaluator design  (Reviewer nFGD, Reviewer ZAqu)**;

**(5) The experimental setting aligns with the real-world application scenario (Reviewer nFGD), thorough analysis and well-support results (Reviewer nFGD, Reviewer 3iA3, Reviewer ZAqu), significant performance improvements (Reviewer jBkz).**

We greatly appreciate the positive comments on our work. These comments encourage us to continue our efforts in advancing this very young research area of GNN model evaluation. We are expecting this work can be beneficial to advance the GNN model inference and deployment in real-world applications.
More detailed responses are as follows. We hope our responses address all weaknesses and questions! Please let us know if there is any concern. We have considered your valuable suggestions, and have modified accordingly to improve the manuscript in the final version.

---

### Decision · Program_Chairs · 2023-09-21

**Decision:**

Accept (poster)

**Comment:**

This paper studies the important problem of evaluating a GNN model, i.e., to estimate the accuracy on unseen unlabeled data for a well trained model. The methodology presented is interesting and inspiring. The paper is well written, the ideas are novel and the experimental results are convincing.